# Effect of the Harvest Season of *Anthyllis henoniana* Stems on Antioxidant and Antimicrobial Activities: Phytochemical Profiling of Their Ethyl Acetate Extracts

**DOI:** 10.3390/molecules28093947

**Published:** 2023-05-07

**Authors:** Amani Ayachi, Amer Ben Younes, Ameni Ben Ammar, Bouthaina Bouzayani, Sonda Samet, Mariam Siala, Mohamed Trigui, Michel Treilhou, Nathan Téné, Raoudha Mezghani-Jarraya

**Affiliations:** 1Laboratory of Organic Chemistry LR17ES08, Natural Substances Team, Faculty of Sciences of Sfax, University of Sfax, P.B 1171, Sfax 3000, Tunisia; amaniayachi21@gmail.com (A.A.); amer.benyounes@live.fr (A.B.Y.); bouzayanibouthaina@yahoo.com (B.B.); samet.sonda95@gmail.com (S.S.); raoudhajarraya@yahoo.fr (R.M.-J.); 2Laboratory of Environmental Sciences and Sustainable Development “LASED”, Sfax Preparatory Institute for Engineering Studies, University of Sfax, P.B 1171, Sfax 3000,Tunisia; ameni.benammar@enis.tn (A.B.A.); mariam.siala@ipeis.usf.tn (M.S.); mohamed.trigui@ipeis.usf.tn (M.T.); 3Equipe BTSB-EA 7417, Institut National Universitaire Jean-François Champollion, Université de Toulouse, Place de Verdun, 81012 Albi, France; michel.treilhou@univ-jfc.fr

**Keywords:** *Anthyllis henoniana*, LC-HESI-MS^n^, ethyl acetate extracts, antioxidant and antimicrobial activities

## Abstract

*Anthyllis henoniana* stems were harvested in two seasons: winter and spring (February and May 2021). In this study, we investigated the antioxidant (DPPH, ABTS, FRAP and TAC) and antimicrobial activities, total phenolic contents and total flavonoid contents of the obtained extracts (hexane, ethyl acetate and methanol). The results showed that ethyl acetate extract from stems harvested in winter exhibited the highest antioxidant activity, while ethyl acetate extract from the stems harvested in spring showed the most potent antibacterial and antifungal activities. To explain these differences, we investigated the phytochemical composition of these two extracts using liquid chromatography coupled with mass spectrometry. Therefore, 45 compounds were detected, from which we identified 20 compounds (flavonoids, triterpenoids, chalcones and phenolic acids); some were specific to the harvest month while others were common for both periods. Some of the major compounds detected in ethyl acetate (spring) were dihydrochalcone (Kanzonol Y, 8.2%) and flavanone (sophoraflavanone G, 5.9%), previously recognized for their antimicrobial effects. We therefore concluded that the difference in activities observed for the two harvest periods depends on the chemical composition of the extracts and the relative abundance of each compound.

## 1. Introduction

Plant metabolites, primarily phenolic compounds, are a valuable resource for humans. The antioxidant activity responsible for neutralizing free radicals, which are harmful to living organisms, is considered as the main property provided by phenolic compounds [1].

These compounds are the most abundant secondary metabolites of plants [2], and they include different families widely distributed in nature and found in most foods of vegetable origin. Their chemical variability includes several compounds from simple phenolic acids to complex structures such as flavonoids. These components are known to be potent due to the presence of an aromatic ring and a hydroxyl group in their structure. This antioxidant effect consequently reduces the risk of diseases [2], and some antioxidants are regarded as antimicrobial [3]. These substances contribute to protecting plants from oxidative damage and can be used as antioxidants by humans [2].

The genus *Anthyllis* includes twenty species growing in Europe, Africa and the Mediterranean Basin. The species *Anthyllis henoniana* was described as a perennial, silky, hairy shrub species 30 to 60 cm tall belonging to the Fabaceae family and comes to its vegetative activity after the first autumn rains. Its blooming phase begins in late winter [4]. Southern Tunisia has an arid Mediterranean climate with a long-term annual rainfall of 80 mm concentrated in the growing season between September and April and a dry season lasting about 4 months from May to August [5]. It was previously reported that the content of secondary plant metabolites is not stable and depends on the growth stage, the part of the plant and the characteristics of the environment. A research study on mistletoe extracts showed that the chemical profile and biological activity of the plant material are related to the climatic conditions [6]. Based on these pieces of information, we tried to provide a phytochemical screening of hexane, ethyl acetate and methanol stem extracts from *A. henoniana* harvested in winter (February 2021) and spring (May 2021) to provide a better understanding of how the harvest season affects the chemical composition as well as the biological activities of plants.

A previous study carried out on *Anthyllis henoniana* flowers showed its strong antioxidant and antidiabetic activities. It proved that its methanol and ethyl acetate extracts contained the highest amounts of total phenolic content and thus presented the strongest radical scavenging activities and total antioxidant capacities [7].

The current work addresses the evaluation of total phenolic and flavonoid contents and the in vitro antioxidant and antimicrobial activities of *A. henoniana* stem extracts harvested in winter and spring. Moreover, based on the obtained results, the chemical composition of both ethyl acetate extracts from the stems harvested in winter and spring was compared using LC-MS/MS analysis. In addition to the antidiabetic and antioxidant activities of *A. henoniana* flowers, there are no specific scientific reports nor specific references dealing with their antifungal activity and the identification of their extracts’ composition using chromatographic techniques.

## 2. Results and Discussions

### 2.1. Total Phenolic and Flavonoid Contents of Various Anthyllis Henoniana Stem Extracts

The total phenolic contents (TPC) and total flavonoid contents (TFC) of *A. henoniana* extracts obtained in winter and spring are determined in Table 1.

Table 1 shows a difference in the phenolic and flavonoid contents of stems collected in winter and spring.

Total phenolic contents (TPC): The results obtained for the stems collected in winter showed that ethyl acetate extract presented the highest significant value (668.36 mg GAE/g E), followed by methanol and hexane extracts, respectively (568.58 and 202.9 mg GAE/g E). On the other hand, the stems collected in spring presented the highest significant content of total phenols with methanol extract (751.78 mg of GAE/g E), followed by ethyl acetate and hexane extracts with values of 567.96 and 198.33 mg GAE/g E, respectively.

Total flavonoid contents (TFC): The results obtained for this parameter followed the same curve as those of TPC. The highest flavonoids concentration was found with ethyl acetate extract when the stems were collected in winter, while methanol extract had the highest significant concentration when the stems were collected in spring.

### 2.2. Evaluation of the Antioxidant Activity of Stem Extracts

Generally, there is no specific method to evaluate and quantify the efficacy of an antioxidant, as there are various reaction mechanisms involved in the process of oxidative stress. Therefore, it is recommended to use several methods to evaluate the efficiency of an antioxidant [8].

In this study, four complementary methods were used to evaluate the antioxidant activity of *A. henoniana* stem extracts harvested in two periods. Vitamin C was chosen as a reference.

#### 2.2.1. Free Radical Scavenging 2,2-Diphenyl-1-picrylhydrazyl (DPPH)

The radical scavenging activity of the six extracts and the standard increased with concentration in a dose-dependent manner. It is noteworthy that the comparison of the necessary amount of each extract to inhibit 50% of free DPPH^.^ radicals showed that they all exhibited IC_50_ values close to vitamin C. (0.03 mg·mL^−1^).

For the stems harvested in spring, the methanol extract presented the highest DPPH^.^ scavenging activity (0.04 mg·mL^−1^) followed by ethyl acetate extract (0.05 mg·mL^−1^). Opposing that, the stems harvested in winter showed that the ethyl acetate extract exhibited the strongest scavenging activity compared to the methanol one. Hexane extracts of both harvesting periods did not reach 50% of DPPH^.^ radical inhibition, indicating their weak antioxidant activity. The AAI values (Table 2) ranging from 1 to 1.3 confirm the strong antioxidant activity and validate the obtained results.

These results can be explained by the presence of phenolic compounds, such as flavonoids in polar extracts that can donate their mobile hydrogen of the hydroxyl group to stabilize the free radicals [9]. Hence, the radical scavenging activity of both methanol and ethyl acetate extracts can be attributed to the presence of phenolic compounds. Structural features of flavonoids can also present an important factor. It was confirmed in previous studies that the 2,3-double bond in conjugation with 4-oxo and 3-hydroxyl groups, as well as the 3′-C hydroxyl group, are considered as the main cause behind the stabilization of phenoxyl radicals by electron delocalization across the aromatic ring [10].

#### 2.2.2. Radical Scavenging Activity on 2,2′-Azinobis (3-ethylbenzothiazoline)-6-sulfonic Acid (ABTS)

An ABTS test was used as a second test in order to validate the results obtained with DPPH. This test is based on the trapping capacity of the proton of the cationic radical ABTS^+^. As shown in Table 3, it is noteworthy that both harvest seasons gave IC_50_ values close to that of the standard TROLOX. As observed with DPPH, the radical scavenging activity of ethyl acetate and methanol in both harvest seasons was the strongest, as they presented the lowest IC_50_ values. Due to their low ability to extract phenolic compounds, hexane extracts did not inhibit 50% of free radicals with this method.

#### 2.2.3. Ferric Reducing Antioxidant Power (FRAP)

This assay is used to evaluate the ability of an antioxidant to reduce a colorless complex (Fe^3+^/tripyridyltriazine) to a blue complex (Fe^2+^/tripyridyltriazine) by donating an electron. Therefore, the reducing power of all extracts and vitamin C (standard) was determined.

For the stems collected in winter, we noticed that at the highest concentration (1 mg·mL^−1^), ethyl acetate extract displayed the strongest reducing power (2.0) compared to methanol extract (1.5). As expected from apolar extracts, hexane exhibited the lowest absorbance values.

Unlike the stems collected in spring, Figure 1 shows that at the highest concentration (1 mg·mL^−1^), methanol extract reduced ferric ions Fe^3+^ to ferrous ions Fe^2+^ more effectively with an absorbance value (2.0) close to the vitamin C one (2.5), followed by ethyl acetate extract (1.45).

#### 2.2.4. Total Antioxidant Activity (TAC)

This assay is based on the reduction of Mo^6+^ to Mo^5+^ by the tested extracts and the formation of green phosphate/Mo^5+^ complex with a maximum absorption at 695 nm. These results (Figure 2) revealed that in the extracts obtained from stems collected in winter, the ethyl acetate extract gave the highest value (511.72 mg GAE/g E), followed by vitamin C (430.25 mg GAE/g E), methanol (422.69 mg GAE/g E) and hexane (218.36 mg GAE/g E) extracts, respectively. As for the extracts obtained from stems collected in spring, the methanol extract once again displayed the sharpest value (656.94 mg GAE/g E), followed by ethyl acetate (545.37 mg GAE/g E), vitamin C (430.25 mg GAE/g E) and hexane (168.38 mg GAE/g E), respectively.

These obtained results support the idea that the presence of phenolic compounds is the main cause behind the antioxidant activity and that TAC is directly related to the presence of secondary metabolites in each extract and depend on their nature, quantity and structure, which act synergistically and increase this activity [11].

#### 2.2.5. Correlations between TFC, TPC and Antioxidant Activity

In order to prove the importance of phytochemical constituents to antioxidant capacity, we determined the correlations between the phenolic contents and antioxidant activity of *A. henoniana* stem extracts.

The obtained results (Table 4) disclosed important correlations for all extracts between the TPC and DPPH and the TFC and DPPH, with correlation coefficients R^2^ = 0.949 and R^2^ = 0.980, respectively. Similarly, an important linear correlation was established between the different phenolic contents of extracts and FRAP, with R^2^ = 0.992 for TPC and R^2^ = 0.986 for TFC. For TAC and the different phenolic contents, we noticed strong linear correlations with the respective coefficients R^2^ = 0.967 (TAC-TPC) and R^2^ = 0.944 (TAC-TFC).

These statistical results indicated the presence of a significant correlation (*p* < 0.05) between FRAP and TAC (R^2^ = 0.935). Similarly, the DPPH test showed a good correlation with FRAP and TAC (R^2^ = 0.952 and R^2^ = 0.900, respectively).

A significant statistical correlation was also observed between the scavenging activity of DPPH and that of the ABTS radical (R^2^ = 0.943). This can be explained by the similar mechanism used in these two methods as they are both based on the scavenging activity of free radicals.

These observations show that antioxidant components present in *A. henoniana* extracts contributed to increasing the antioxidant activity.

### 2.3. Evaluation of the Antimicrobial Activity of Stem Extracts

The in vitro antibacterial and antifungal activities of *A. henoniana* stem extracts, prepared from the plants harvested in winter and spring (February and May 2021), were tested qualitatively and quantitatively by the presence or absence of inhibition zones toward a panel of microorganisms and quantitatively by the determination of the MIC and MBC for bacterial strains and MFC for fungal strains.

The microorganisms tested in the present investigation are known as opportunists for humans and animals and cause food contamination and deterioration.

The active extracts were then quantitatively valued by the determination of their MIC, MBC and MFC values.

#### 2.3.1. Inhibition Zones, MIC, MBC and MFC

Among the tested extracts, the ethyl acetate from the plant harvested in spring exhibited antimicrobial activity against 70% of the tested strains. The inhibition zones were in the range of 8–22 mm with MIC values of 0.625–10 mg·mL^−1^ for the Gram-positive bacteria and 5–10 mg·mL^−1^ for the Gram-negative ones (Table 5 and Table 6). However, *A. henoniana* ethyl acetate stem extract prepared from the plant harvested in winter exhibited activity against only 46% of the tested microorganisms with low inhibition zones and high MIC and MBC values. The data obtained from this study demonstrated more susceptibility of the Gram-positive bacteria than Gram-negative ones to the extract. This susceptibility spring is due to the complexity of the double membrane-containing cell envelope in Gram-negative bacteria compared to the single membrane structure of positive ones [12]. The antimicrobial activity of extracts could be due to the disruption of the permeability barrier of cell membrane structures and the loss of chemiosmotic control [13]. This study also revealed that *A. henoniana* bioactivity is dependent on the season since these extracts are collected in February and May, corresponding to winter and spring in Tunisia, respectively.

The strong antimicrobial activity of the *A. henoniana* ethyl acetate stem extract from the plant harvested in spring correlated positively with the richness of this extract in phenolic and flavonoid compounds, reported to inhibit the multiplication of pathogens [14,15]. From these results, *A. henoniana* ethyl acetate stem extract could be exploited as a natural target against pathogens in food and medical industries. As with the antimicrobial activity, we have evaluated the antifungal activity by means of inhibition zone diameters compared to the positive control, Amphotericin B. Selected fungal strains, such as *F. oxysporum*, *B. cinerea* and *A. niger*, are known for their mycotoxins being considered a challenge for agriculture and food industries. The tested *A. henoniana* extract exhibited strong antifungal activity with an inhibition zone in the range of 9–19 mm and MIC of 1.125 to 2.5 mg·mL^−1^ against *F. oxysporum* and *A. alternata* and MFC values of 4.5 and 9 mg·mL^−1^. No activity was detected against *B. cinerea* at the tested concentration.

The ratio of minimum bactericidal/fungicidal concentration (MBC/MFC) to minimum inhibitory concentration (MIC) provides information regarding the degree of bactericidal or fungicidal action of the drugs. Antimicrobial substances are usually regarded as bactericidal or fungicidal if the MBC/MIC or MFC/MIC ratio is less than or equal to four. If the ratio is higher than four, the extract is bacteriostatic/fungistatic. Therefore, it is possible to obtain drug concentrations that will kill 99.9% of the organisms exposed. Based on this classification and the obtained experimental data in this study, it can be concluded that the *A. henoniana* ethyl acetate extract (May) showed bactericidal and fungicidal effects for most tested microorganisms [16].

#### 2.3.2. Correlations among Phenolic Contents and Inhibition Zones Diameter

As we previously stated, among all tested extracts, the ethyl acetate one from the plant harvested in spring exhibited the strongest antimicrobial activity. Therefore, we proceeded to study the correlations between phenolic contents and inhibition zones for this extract. The correlation between antimicrobial activity (bacterial and fungal strains) and phenolic contents (TPC and TFC) is shown in Table 7 (*p* < 0.05). The outcomes showed important positive and negative correlations. TPC and TFC appeared to have a positive correlation, suggesting that these two variables move in the same direction and influence each other. TPC appears to positively and negatively correlate with some of the Gram-positive bacteria (*B.subtilis* and *S.epidermis)* as well as the Gram-negative bacteria (*M. luteus* and *K. pneumoniae)*. Regarding TFC, the obtained data also revealed correlations with Gram-positive bacteria (*S. aureus, S. epidermis, E. faecalis* and *M. luteus*), Gram-negative bacteria (*S. enteritidis)* and fungi (*F. oxysporum).* These observations confirm that phenolic compounds are the main reason behind the antimicrobial activity of plant extracts [17].

### 2.4. LC-MS-MS

The results of the antioxidant activity, TPC, TFC and antimicrobial effect showed that ethyl acetate extracts can act differently according to the harvesting period. To explain these observations, we investigated both extracts (winter and spring) using HPLC coupled with hot electrospray ionization mass spectrometry in negative mode. The obtained mass chromatograms are illustrated in Figure 3. For each peak, we indicated the retention time, the relative abundance, the UV, the deprotonated mass and the fragment ions generated by MS^2^ and MS^3^ (Table 8).

#### 2.4.1. Characterization of the Detected Compounds

The analysis of chromatograms A and B shows several similarities and some slight differences. The notable differences between the two extracts are especially in the relative abundance of the detected compounds. We have identified 20 compounds classified in several families such as flavonoids, phenolic acids and triterpenoids.

Compound **1** (T_R_ = 9.89 min) showed a molecular ion at *m/z* 593. Its UV spectra (Table 8) showed characteristic absorption bands of flavonoids ranging between 240 and 285 nm and 300 and 385 nm [36]. It presented characteristic fragment ions at 575 [(M − H)-18] (dehydration), 503 [(M − H)-90] (loss of C_3_H_6_O_3_ from the sugar unit), 473 [(M − H)-30] (loss of CHOH), 383 [(M − H)-90] (loss of C_3_H_6_O_3_ from the sugar unit) and 353 [(M − H)-30] (loss of CHOH). This compound was identified as vicenin-2 [18].

Compound **2** (T_R_ = 10.47 min) generated a molecular ion at *m/z* 563. Its UV spectra presented characteristic absorption bands of flavonoids ranging between 282 and 283 nm. Its MS^2^ mode generated a fragmentation of a hexose and gave product ions 545 [(M − H)-18] (dehydration), 503 [(M − H)-42] (loss of C_2_H_2_O), 473 [(M − H)-30] (loss of CH_2_O) and 443 [(M − H)-30] (loss of CH_2_O). The ions 383 [(M − H)-60] and 353 [(M − H)-30] observed in both MS^2^ and MS^3^ are the result of the fragmentation of the pentose. This compound was identified as apigenin-6-*C*-glucoside-8-*C*-arabinoside (schaftoside) [19].

Compound **3** (T_R_ = 11.14 min) generated a molecular ion at *m/z* 533 which can be attributed to apigenin-*C*-pentoside-*C*-pentoside. The loss of a rhamnoside resulted in the apparition of a major MS^2^ fragment at *m/z* 443 [(M − H)-90] due to the fragmentation of the pentoside [20].

Compound **4** (T_R_ = 12.32 min) exhibited a molecular ion at *m/z* 769. The loss of a rhamnoside resulted in the apparition of a major MS^2^ fragment at *m/z* 623 [(M − H)-146]. The MS^3^ method generated *m/z* 357 [(M − H)-266] after the loss of C_10_H_18_O_8_, a characteristic fragment of a deprotonated isorhamnetin at *m/z* 315 [(M − H)-42] (loss of C_2_H_2_O), an *m/z* 300 [(M − H)-15] (loss of CH_3_) and an *m/z* 271 [(M − H)-29] (loss of CHO). Therefore, this compound was identified as isorhamnetin glucosyl-di-rhamnoside [21].

Compound **5** (T_R_ = 13.04 min) presented a molecular ion at *m/z* 623 with product ions at *m/z* 315 [(M − H)-308] (loss of disaccharide), *m/z* 300 [(M − H)-15] (loss of CH3) and *m/z* 271 [(M − H)-29] (loss of CH_2_O) and was identified as isorhamnetin-3-*O*-rutinoside [22].

Compound **6** (T_R_ = 14.33 min) showed a molecular ion at *m/z* 373. The obtained MS^2^ fragments were in good agreement with those of 7-Hydroxymatairesinol reported in the literature [23].

Compound **7** (T_R_ = 14.66 min) showed a molecular ion at *m/z* 473. The major ion generated at *m/z* 455 obtained in MS^2^ mode is the result of dehydration [24]. This compound was identified as an Asiatic acid/madecassic acid derivative.

Compound **8** (T_R_ = 15.92 min) at *m/z* 399 showed a prominent ion at *m/z* 381 after dehydration, suggesting the presence of a compound of benzofurane type [25].

Compound **9** (T_R_ = 16.18 min) at *m/z* 457 was identified as lucidenic acid A, taking into account the provided data from the literature. The MS^2^ showed a major fragment at *m/z* 439 due to dehydration. A second dehydration gave the fragment ion at *m/z* 421 [26].

Compound **10** (T_R_ = 16.45 min) showed a molecular ion at *m/z* 371 and was identified as a caffeoyl glucarate isomer. The major characteristic ion generated at *m/z* 353 obtained in MS^2^ mode is the result of dehydration [27].

Compound **13** (T_R_ = 19.00 min) showed a molecular ion at *m/z* 313. The major characteristic ion generated at *m/z* 298 obtained in MS^2^ mode is the result of demethylation and was identified as hedysarimpterocarpene A (HPA) [25].

Compound **14** (T_R_ = 19.27 min) showed a molecular ion at *m/z* 355 and can be identified as chebulic acid. The MS^2^ mode generated a major ion at *m/z* 337 [(M − H)-18] (dehydration). As for the MS^3^ mode, it presented a fragment ion at *m/z* 305 [(M − H)-32] (loss of O_2_) and a major ion at *m/z* 165 [(M − H)-140] (C_7_H_8_O_3_) [28].

Compound **15** (T_R_ = 20.51 min) showed a molecular ion at *m/z* 439 and can be identified as prenylated licoriphenone. The MS^2^ mode generated a major ion at *m/z* 421 [(M − H)-18] (dehydration) [29].

Compound **16** (T_R_ = 21.25 min) presented a deprotonated ion at *m/z* 341 and gave a major fragment at *m/z* 323 after the loss of H_2_O (dehydration). The fragment at *m/z* 297 appeared due to the loss of CO_2_. The fragment ion at *m/z* 151 [(M − H)-118] (loss of C_7_H_2_O_2_) is the result of a rupture of the liaison 1→3 from the sugar unit. These fragments appear to be the same as those found in caffeic acid-*O*-glycoside [30].

Compound **27** (T_R_ = 29.63 min) showed a molecular ion at *m/z* 425 and was identified as abscisic acid hexoside. The MS^2^ mode generated fragment ions at *m/z* 407 [(M − H)-18] (dehydration), *m/z* 353 [(M − H)-54] (loss of C_3_H_2_O) and a major *m/z* 219 [(M − H)-90] (loss of a sugar moiety) [31].

Compound **28** (T_R_ = 30.57 min) showed a molecular ion at *m/z* 421 and was identified as 6,8-diprenylkaempferol. The MS^2^ mode generated the same fragmentation pattern shown in the literature [32].

Compound **29** (T_R_ = 31.35 min) and compound **31** (T_R_ = 32.55 min) generated a common molecular ion *m/z* (423) and an identical fragmentation pattern and were assigned to sophoraflavanone G. The difference in their retention time suggests the possibility of two isomers. The MS^2^ mode generated a major fragment ion at *m/z* 405 due to the loss of H_2_O [33].

Compound **34** (T_R_ = 35.62 min) generated a molecular ion at *m/z* 409 and was attributed to a dihydrochalcone known as Kanzonol Y. The MS^2^ mode presented a characteristic of the proposed compound at *m/z* 391 [(M − H)-18] (loss of H_2_O) [34].

Compound **41** (T_R_ = 46.83 min) showed a molecular ion at *m/z* 391 and was attributed to hispaglabridin A. The MS^2^ mode generated a major fragment ion at *m/z* 203 [(M − H)-188] (loss of C_12_H_12_O_2_), *m/z* 187 due to the loss of an oxygen radical and an *m/z* 159 after the loss of COH [35].

#### 2.4.2. Findings Interpretation

The obtained results showed that the harvest month can affect the relative abundance of each detected compound’s obtained extracts as well as the antioxidant and antimicrobial activities. We found that the ethyl acetate extract of the stems collected in winter has a high antioxidant activity compared to the one obtained from the stems collected in spring. The opposite was observed for the antimicrobial activity, as the ethyl acetate extract obtained in spring was more active.

Previous research studies reported that the additive synergetic effect of an extract could be more effective than the separated compounds [8]. Therefore, the obtained results suggest that the compounds found in stems harvested in winter could act synergistically as strong antioxidants but not as antimicrobials.

Previous studies reported that variation in the phytochemical composition and antioxidant and antimicrobial activities could be explained by ecological factors, type of extraction solvents, seasons and the class of phytochemicals present. Furthermore, environmental factors can highly affect the quantity of bioactive compounds [37]. Research works also previously concluded that differences in the antioxidant activity harvested from different trees and in different seasons can be attributed to factors such as climate and temperature, which can significantly affect the chemical composition and antioxidant activity of plants. It was also considered that the chemical profile and the biological activity of plant material are related to climatic conditions [6].

Major compounds in each extract can also be an important factor. For example, the major compound (*m/z* 393) detected in both extracts could influence the observed results. This compound presented a higher relative abundance in spring (14.7%) compared to winter (12.2%), so its presence could influence the antimicrobial effect of the extract. There were no literature data regarding its structure. Other major compounds detected in ethyl acetate (spring) were dihydrochalcones (Kanzonol Y, 8.2%) and flavanones (sophoraflavanone G, 5.9%), which have previously been recognized for their antimicrobial effects [38]. Previous studies reported that dihydrochalcones acted more effectively against Gram (+) strains such as *S. aureus* than Gram (−) strains [39]. Daseul et al. proved that sophoraflavanone G is also known for its strong antimicrobial activity against Gram (+) bacteria as it disturbs bacterial cell walls by binding with PGN (the major component of the cell wall of Gram-positive bacteria) [40]. Sakar et al. proved that Gram-positive bacteria are more sensitive than Gram-negative bacteria to the action of plant extracts. The main cause behind this sensitivity is the presence of an outer membrane that contains hydrophilic lipopolysaccharides encompassing the bacterial peptidoglycan layer in Gram-negative bacteria that acts as a barrier for macromolecules as well as hydrophobic compounds, thus limiting the diffusion of hydrophobic compounds into the bacterium’s cytoplasm [41]. This is in good agreement with the antimicrobial effect caused by ethyl acetate extract (spring), as it acted more effectively against Gram (+) strains.

## 3. Materials and Methods

### 3.1. Collection and Extraction of Plant Material

*Anthyllis henoniana* was collected in winter (February 2021) and spring (May 2021) from the Sahara of Beni Khedache, Medenine, Tunisia, approximately to this GPS coordination: 33°13′09.6″ N 10°12′46.2″ E. The climate of this area is arid Mediterranean with a rainy season concentrated during autumn and a dry season starting from May. The area is known for its sandy soil and sand accumulation as well as the presence of limestone. Dr. Zouhair Bouallagui performed the botanical identification in the botany laboratory of the Faculty of Sciences, University of Sfax, Tunisia, and a voucher specimen (LCO 140) was deposited at the herbarium of the Laboratory of Organic Chemistry (LR17-ES08), Natural Substances Team, Faculty of Sciences, University of Sfax. An amount of 500 g of stems (from each harvest period) was dried, milled, placed in the cotton cartridge and extracted successively with hexane, ethyl acetate and methanol using a Soxhlet extractor. All the extracts were then filtered, evaporated using a rotavapor at 40 °C and stored at 4 °C prior to analysis.

### 3.2. Total Phenolic and Total Flavonoid Contents

#### 3.2.1. Total Phenolic Contents (TPC)

To quantify the total hydroxyl groups in the obtained extracts, the total phenolic contents (TPC) of *Anthyllis henoniana* stem extracts were determined using the Folin–Ciocalteu method presented by Ben Younes et al. [7]. In total, 0.5 mL of Folin–Ciocalteu reagent and 5 mL of Na_2_CO_3_ (20%) were added to 0.1 mL (1 mg·mL^−1^) of each extract. After 30 min, the absorbance of each mixture was measured at 727 nm. Gallic acid was used as a standard. The phenolic contents in the extracts were expressed in terms of gallic acid equivalent per gram of dried extract (mg of GAE/g E). The six extracts were analyzed in triplicate.

#### 3.2.2. Total Flavonoid Contents (TFC)

The total flavonoid contents of *Anthyllis henoniana* stem extracts were identified by the aluminum chloride method. We added 0.3 mL of NaNO_2_ (5%) and 4 mL of water to 1 mL of each extract. The mixture was allowed to stand for 5 min, followed by the addition of 0.3 mL of AlCl_3_ (10%) and then 2 mL of NaOH (1 M). The mixture was adjusted to 10 mL with water. Absorbance was measured at 510 nm after 15 min of incubation at room temperature. The extracts were analyzed in triplicate, and TFC was measured in milligrams of quercetin equivalent (QE) per gram of extract [7].

### 3.3. Antioxidant Assay

The divergence of antioxidant capacity could not fully be described by a single method. Thus, it should be determined by several assays using the same initial concentration such as free radical scavenging 2,2-diphenyl-1-picrylhydrazyl (DPPH), radical scavenging activity on 2,2′-azinobis (3-ethylbenzothiazoline)-6-sulfonic acid (ABTS), ferric reducing power (FRAP) and total antioxidant capacity (TAC). For each assay, the obtained extracts were tested at different concentrations (0.0625, 0.125, 0.25, 0.5 and 1 mg·mL^−1^).

#### 3.3.1. Free Radical Scavenging 2,2-Diphenyl-1-picrylhydrazyl (DPPH)

The antiradical activity of the extracts was evaluated using the 2,2′-diphenyl−1-picrylhydrazyl (DPPH). Briefly, 1 mL of each extract concentration was added to 2 mL of ethanolic solution of DPPH (0.04 g·L^−1^). The tubes were incubated in the dark for 30 min at room temperature; the absorbance was taken at 517 nm. The experimental results were then compared with the control that contained 1 mL of 95% ethanol and vitamin C solution [42]. The antioxidant concentration reducing 50% of DPPH free radicals (IC_50_) was then determined for each extract. The findings were expressed using the inhibition percentage and the antioxidant activity index formula:PI (%) = [(A _control_ − A _sample_)/A c_ontrol_] × 100
AAI = Final concentration of DPPH (µg·mL^−1^)/IC_50_ (µg·mL^−1^)
A_control_: the negative control absorbance
A_sample_: the sample absorbance.

On the report of AAI values, antioxidant activity is considered poor when AAI < 0.5, moderate when 0.5 ≤ AAI ≤ 1, strong when 1.0 ≤ AAI ≤ 2.0 and very strong when AAI >2.0 [42].

#### 3.3.2. Radical Scavenging Activity on 2,2′-Azinobis (3-ethylbenzothiazoline)-6-sulfonic Acid (ABTS)

An ABTS stock solution (7 mM in water) was mixed with 2.45 mM potassium persulfate to obtain ABTS^•+^. The mixture was then incubated for 12 to 16 h in the dark at room temperature in order to reach stable oxidative stress. The ABTS^•+^ solution was diluted with PBS (pH 7.4) to an absorbance of 0.700 at 734 nm. For the spectrophotometric assay, 3 mL of the ABTS^•+^ solution and 20 μL of standard (TROLOX) or each extract were mixed, and the absorbance was determined at 734 nm at 1,6, 10, 60, 120, 180, 360 and 1440 min after mixing [43].
ABTS^•+^ radicals PI (%) = [(DO_b_ − DO_a_)/DO_b_] × 100

Here, DO_b_ refers to the absorbance of the control (without extract) and DO_a_ to the absorbance of the sample (with extract).

#### 3.3.3. Ferric Reducing Antioxidant Power (FRAP)

As described by Affes et al. [9], the reducing power of the obtained extracts was determined by dissolving 1 mg stem extracts in 1 mL of ethanol, which was then mixed with 2.5 mL of potassium ferricyanide [K_3_Fe(CN_6_)] (1%, *w/v*) and 2.5 mL of phosphate buffer (0.2 M). The tubes were then incubated for 20 min at 50 °C, and then 2.5 mL of a trichloroacetic acid solution (10% *w/v*) was added to the mixture, followed by centrifugation of the samples at 3000 rpm for 10 min. As a final step, 2.5 mL of the supernatant solution was mixed with 0.5 mL of the 0.1% (*w/v*) solution of ferric chloride (FeCl_3_) and 2.5 mL of distilled water. The absorbance was measured at 700 nm, and vitamin C was chosen as the standard. Therefore, the observed increase in absorbance of *Anthyllis henoniana* stem extracts displayed the reducing power.

#### 3.3.4. Total Antioxidant Capacity (TAC)

Using the formation of the complex phosphomolybdenum as described by Affes et al. [9], this assay shows the reduction in ammonium molybdate by resulting in a green ammonium phosphate/molybdate complex. An aliquot of 0.1 mL of a sample solution containing each extract was combined with 1 mL of reagent solution (0.6 M sulfuric acid, 28 mM sodium phosphate and 4 mM ammonium molybdate). The testing solution was incubated in a water bath at 95 °C for 90 min. After cooling at room temperature, the absorbance was measured at 695 nm against a blank containing 1 mL of reagent solution in which the extract had been replaced with the appropriate volume of the same solvent used for the sample. Gallic acid was used as standard in the assay.

### 3.4. Antimicrobial Activity

#### 3.4.1. Microbial Strains

Nine bacterial strains including Gram-positive and Gram-negative bacteria were used to investigate the antibacterial activity of the extracts. These bacteria were obtained from the international culture collections (ATCC) and the local culture Collection of Tunisian Microorganisms “CTM” of the Center of Biotechnology of Sfax. The Gram-positive bacteria are *Bacillus subtilis* (ATCC 6633), *Staphylococcus aureus* (ATCC 25923), *Enterococcus faecalis* ATCC 29212, *Micrococcus luteus* (ATCC 1880) and *Staphylococcus epidermis* (ATCC 12228). The tested Gram-negative bacteria are *Salmonella enteritidis* (CTM 2133), *Escherichia coli* (ATCC 25922), *Pseudomonas aeruginosa* (ATCC 9027) and *Klebsiella pneumoniae* (ATCC 10031). The bacterial strains were grown on Mueller–Hinton broth (Bio-Rad, France) at 37 °C for 12–14 h. For the antifungal activity, four phytopathogenic fungi were used as indicator strains to evaluate the inhibitory potential of extracts: *Altenaria alternata* (CTM10230), *Fusarium oxysporum* (CTM10402), *Aspergillus niger* (CTM 10099) and *Botrytis cinerea* (LBPES15). All fungi strains were grown on Potato Dextrose Agar (PDA), and plates were incubated for 5–7 days at 28 °C.

#### 3.4.2. Well-Diffusion Agar Assay, Minimum Inhibitory Concentrations (MIC) and Minimum Bactericidal Concentrations (MBC)

Antimicrobial testing was performed by agar well diffusion method as described by Tagg et al. [44] and broth microdilution assay using sterile Mueller–Hinton medium (Bio-Rad, France) for bacterial strains and Potato Dextrose Agar (PDA) for fungal strains. For the antibacterial assays, a fresh cell suspension (100 µL) adjusted to 10^7^ CFU/L was inoculated onto the surface of agar plates. Thereafter, 6 mm diameter wells were punched in the inoculated agar medium with sterile Pasteur pipettes, and the extracts (2.5 mg/well) were added to each well. Negative controls consisted of 20% DMSO and 50% ethanol used to dissolve the plant extracts, and Gentamicin (15 µg/well) was used as a positive control to determine the sensitivity of each bacterial strain, while Amphotericin B (20 µg/well) was used as a positive control for fungal strains. The plate was allowed to stand for 2 h at 4 °C to permit the diffusion of the extracts followed by incubation at 37 °C for 24 h. The antimicrobial activity was evaluated by measuring the inhibition zones (clear zone around the well) against the tested microorganisms. All tests were repeated three times.

Minimum inhibitory concentrations (MIC) of *Anthyllis henoniana’s* extracts were determined according to Eloff in sterile 96-well microplates with a final volume in each microplate well of 200 µL [45]. A two-fold serial dilution of each extract was prepared in the microplate wells over the range 0.0048–10 mg·mL^−1^. To each well, 10 µL of cell suspension was added to the final inoculum concentration of 10^6^ CFU/L. The plates were then covered with sterile plate covers and incubated at 37 °C for 24 h. Gentamicin and Amphotericin B were used as positive drug controls. The MIC was defined as the lowest concentration of the extract at which the microorganism does not demonstrate visible growth after incubation. As an indicator of bacterial growth, 25 µL of 0.5 mg·mL^−1^ *p*-iodonitrotetrazolium chloride (INT), dissolved in sterile water, was added to the wells and incubated at 37 °C for 30 min. The lowest concentration of each sample showing no growth was taken as its minimal inhibitory concentration MIC. For the determination of the minimum bactericidal/fungicidal concentrations (MBC/MFC), 5 µL from each well that showed no change in color was plated on nutrient agar and incubated at 37 °C for 24 h. The lowest concentration that yielded no growth after this sub-culturing was taken as the MBC, indicating that >99.9% of the original inoculum was killed.

### 3.5. LC–MS Analysis

The ethyl acetate extracts of *Anthyllis henoniana* were investigated using a Thermo Scientific LTQ XL Mass Spectrometer (Thermofisher, Courtaboeuf, France) fitted with an electrospray ionization source in the negative mode. Thermo Roadmap software *Xcalibur* 4.4.16.14 was used to record ion spectra. A C_18_ reversed phase Acclaim column at 30 °C (5 µm, 150 mm × 2.1 mm) was delivered to Vanquish HPLC for analysis. A: 0.1% formic acid and 5% acetonitrile in water (*v/v*) and B: 0.1% formic acid in acetonitrile (*v/v*) were the selected solvents. The elution gradient was set starting with 0% to 40% of B for 10 min and then adding 10% of B every 10 min until reaching 80% of B. Lastly, from 80% to 100% of solvent B was added over 3 min and maintained for 2 min before returning to initial conditions. The mobile phase had a flow rate of 250 µL·min^−1^, and the injection volume was 10 µL. High-purity nitrogen served as the nebulizer and auxiliary gas for the HESI source, the ion spray voltage was fixed at 3.5 V, the capillary temperature was calibrated at 300 °C and the sheath and auxiliary gas pressures were set to 50 and 5 psi, respectively. The spectral range was from *m/z* 50 to 1200.

The approach combined full scans and MS^n^ experiments using collision energy ranging from 10 to 35 eV, depending on the molecular mass of compounds.

### 3.6. Statistical Analysis

Replicate errors were in all cases < 10% (n = 3). The differences were analyzed using Duncan and Tukey’s post hoc tests for multiple comparisons with *p* < 0.05. The Statistical Product and Service Solutions program (SPSS) version 20 was used to analyze the differences and calculate the correlation coefficients R^2^ in order to highlight, on the one hand, the correlation between the different phenolic contents of all extracts and their antioxidant and antimicrobial activities and the correlation between the different antioxidant activity tests, on the other.

## 4. Conclusions

The present paper is the first attempt to identify the phenolic compounds from *A. henoniana* stem extracts as well as their antioxidant and antimicrobial activities. We concluded that the harvest period can affect the total phenolic contents (TPC), total flavonoid contents (TFC) and antioxidant activity, as well as the antimicrobial effect. Ethyl acetate of the stems harvested in spring demonstrated strong inhibition of Gram-positive and some Gram-negative bacterial strains. Significant antifungal activity against *Fusarium oxysporum* and *Alternaria alternata* was also detected. Therefore, a determination of the chemical composition using an adapted LC-ESI-MS/MS method of both ethyl acetate extracts was proceeded. The phenolic compounds detected were classified and compared in both extracts. Total phenolic contents and total flavonoid contents as well as the strong antioxidant and antimicrobial activities are the results of the presence of these phenolic compounds. From this perspective, it is noteworthy that our research is a step that can be built upon and taken further, as *A. henoniana* stems had not been extensively studied from the chemical and biological points of view. Our research can expand the potential applications of *A. henoniana* and further provide references for the development and utilization of this plant in pharmaceutical use.

## Figures and Tables

**Figure 1 molecules-28-03947-f001:**
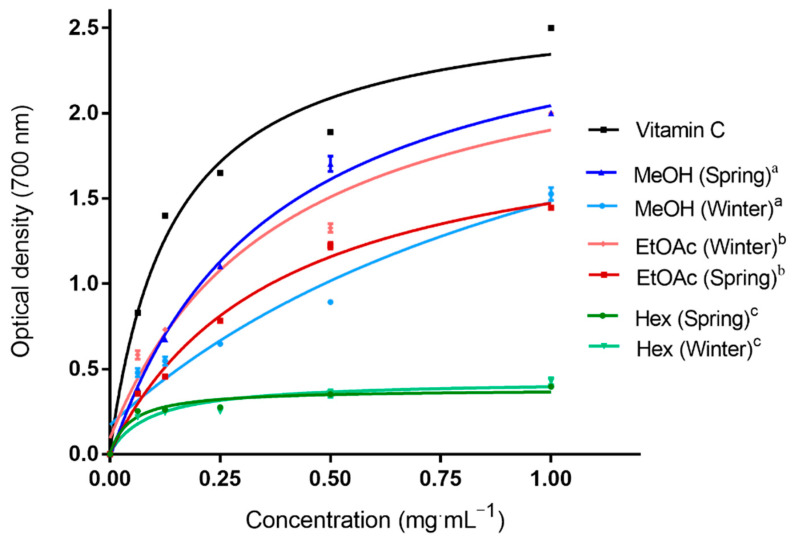
Ferric reducing antioxidant power (FRAP) assays of *A. henoniana* stem extracts and vitamin C. Hex: hexane; EtOAc: ethyl acetate; MeOH: methanol. The differences were analyzed using Duncan and Tukey’s post hoc tests for multiple comparisons with *p* < 0.05: ^a^: strong significance; ^b^: high significance; ^c^: modest significance.

**Figure 2 molecules-28-03947-f002:**
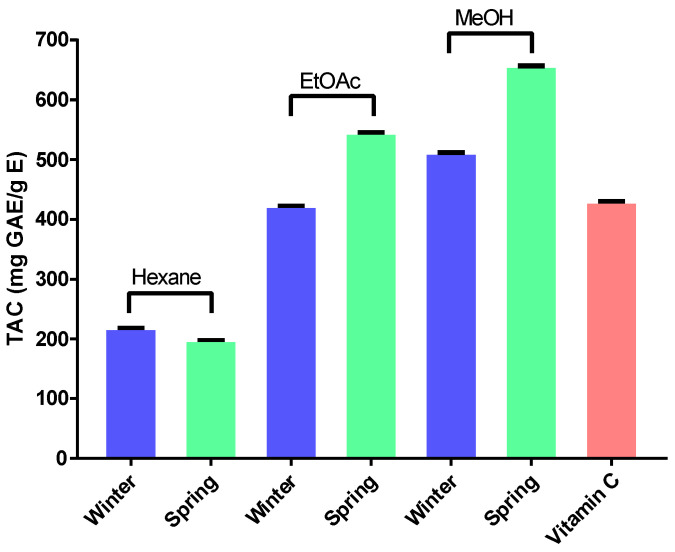
Total antioxidant capacity (TAC) of *A. henoniana* stem extracts compared to vitamin C as a standard. EtOAc: ethyl acetate; MeOH: methanol.

**Figure 3 molecules-28-03947-f003:**
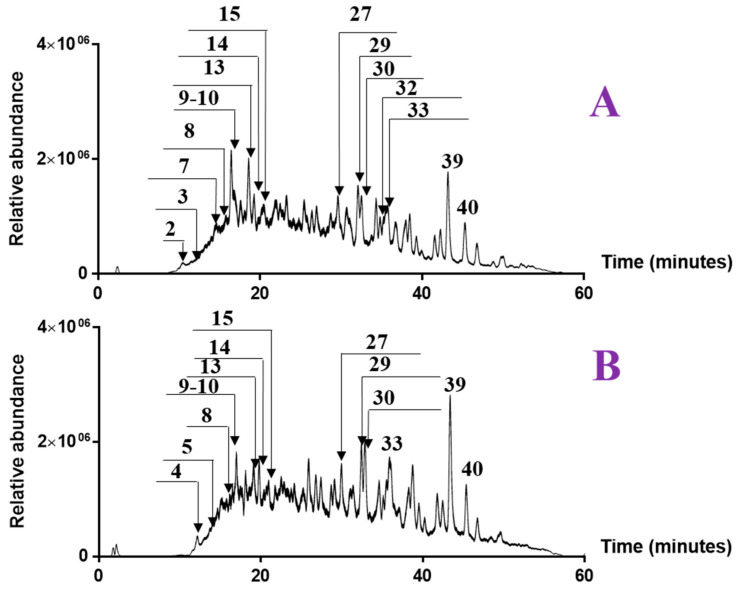
LC-MS chromatograms of ethyl acetate extracts of *A. henoniana* stems; (**A**) stems collected in winter (February 2021); (**B**) stems collected in spring (May 2021).

**Table 1 molecules-28-03947-t001:** TPC and TFC of *A. henoniana* stem extracts harvested in winter and spring.

Harvest Season	Extracts	TPC (mg GAE/g E)	TFC (mg QE/g E)
**Winter**	Hexane	202.96 ± 0.70 ^c^	177.22 ± 0.44 ^c^
	Ethyl acetate	668.36 ± 0.67 ^a^	529.61 ± 0.92 ^a^
	Methanol	568.58 ± 0.53 ^b^	402.73 ± 0.68 ^b^
**Spring**	Hexane	198.33 ± 0.71 ^c^	103.25 ± 0.44 ^c^
	Ethyl acetate	567.96 ± 0.82 ^b^	435.09 ± 0.68 ^b^
	Methanol	751.77 ± 0.67 ^a^	514.20 ± 0.92 ^a^

TPC: total phenolic contents; TFC: total flavonoid contents; GAE: gallic acid equivalent; QE: quercetin equivalent; E: dried extract. Values expressed are means ± S.D. (n = 3). The differences were analyzed using Duncan and Tukey’s post hoc tests for multiple comparisons with *p* < 0.05: ^a^: strong significance; ^b^: modest significance; ^c^: low significance.

**Table 2 molecules-28-03947-t002:** IC_50_ and AAI values of *A. henoniana* stem extracts and vitamin C.

Harvest Season	Extracts	IC_50_ (mg·mL^−1^)	AAI
**Winter**	Hexane	-	-
	Ethyl acetate	0.038 ± 0.002 ^a^	1.01
	Methanol	0.040 ± 0.001 ^b^	1.0
**Spring**	Hexane	-	-
	Ethyl acetate	0.050 ± 0.004 ^b^	0.8
	Methanol	0.040 ± 0.002 ^a^	1.0
**Standard**	Vitamin C	0.030 ± 0.001	1.3

AAI: antioxidant activity index; -: does not reach 50% of inhibition. Values expressed are means ± S.D (n = 3). The differences were analyzed using Duncan and Tukey’s post hoc tests for multiple comparisons with *p* < 0.05: ^a^: strong significance; ^b^: modest significance.

**Table 3 molecules-28-03947-t003:** IC_50_ values of *A. henoniana* stem extracts and TROLOX.

Harvest Season	Extracts	IC_50_ (mg·mL^−1^)
**Winter**	Hexane	-
	Ethyl acetate	0.049 ± 0.001 ^a^
	Methanol	0.053 ± 0.003 ^b^
**Spring**	Hexane	-
	Ethyl acetate	0.059 ± 0.004 ^b^
	Methanol	0.051 ± 0.002 ^a^
**Standard**	TROLOX	0.047 ± 0.001

-: does not reach 50% of inhibition. Values expressed are means ± S.D (n = 3). The differences were analyzed using Duncan and Tukey’s post hoc tests for multiple comparisons with *p* < 0.05: ^a^: strong significance; ^b^: modest significance.

**Table 4 molecules-28-03947-t004:** Correlations among phenolic compounds and antioxidant assays *.

Variables	TPC	TFC	DPPH	FRAP	TAC	ABTS
TPC	1					
TFC	0.981 **	1				
DPPH	0.949 **	0.980 **	1			
FRAP	0.992 **	0.986 **	0.952 **	1		
TAC	0.967 **	0.944 **	0.900 **	0.935 **	1	
ABTS	0.941 **	0.950 **	0.943 **	0.922 **	0.939 **	1

**: the correlation is significant at the 0.01 level. *: Data show the Pearson correlation coefficients (R^2^) between the parameters (*p* < 0.05). TPC: total phenolic contents; TFC: total flavonoid contents; DPPH^•^: antioxidant capacity based on the DPPH^•^ assay; ABTS^•+^: antioxidant capacity based on the ABTS^•+^ assay; FRAP: ferric reducing power; TAC: total antioxidant capacity.

**Table 5 molecules-28-03947-t005:** Inhibition diameter zones in millimeters (mm) of *Anthyllis henoniana* stem extracts against bacterial and fungal strains.

	Harvest Season	IZ (mm) Winter			IZ (mm) Spring			IZ (mm) Control
Microorganisms		Hexane	EtOAc	MeOH	Hexane	EtOAc	MeOH	Gentamicin ^(c)^
**Gram-positive**								
*B. subtilis*		7 ± 0.5 ^a^	9 ± 0 ^a^	11 ± 0.5 ^a^	0 ^b^	17 ± 0.3 ^a^	0 ^b^	20.5 ± 0.2 ^a^
*S. aureus*		0 ^b^	0 ^b^	0 ^b^	0 ^b^	14 ± 0.5 ^a^	12 ± 0.5 ^a^	25.5 ± 1.1 ^a^
*S. epidermidis*		11 ± 0.5 ^a^	9 ± 0.5 ^a^	12 ± 0.5 ^a^	0 ^b^	22 ± 0.6 ^a^	14 ± 0 ^a^	12 ± 0.2 ^a^
*E. faecalis*		0 ^b^	0 ^b^	0 ^b^	0 ^b^	9 ± 0.3 ^a^	0 ^b^	20 ± 0.2 ^a^
*M. luteus*		0 ^b^	0 ^b^	0 ^b^	0 ^b^	10 ± 0.5 ^a^	0 ^b^	20 ± 0.7 ^a^
**Gram-negative**								
*E. coli*		0 ^b^	7 ± 0 ^a^	0 ^b^	0 ^b^	0 ^b^	0 ^b^	21 ± 1.2 ^a^
*P. aeruginosa*		0 ^b^	0 ^b^	0 ^b^	0 ^b^	0 ^b^	07± 0.33 ^a^	18 ± 0.6 ^a^
*K. pneumoniae*		0 ^b^	0 ^b^	0 ^b^	0 ^b^	14 ± 0.5 ^a^	0 ^b^	12 ± 0.5 ^a^
*S. enteritidi* *s*		0 ^b^	0 ^b^	0 ^b^	0 ^b^	8 ± 0.3 ^a^	0 ^b^	18 ± 0.8 ^a^
				**Fungi**				**Amphotericin B ^(d)^ (µg·mL^−1^)**
*B. cinerea*		0 ^b^	0 ^b^	0 ^b^	0 ^b^	0 ^b^	0 ^b^	11.5 ± 0.5 ^a^
*F. oxysporum*		14 ± 0.5 ^a^	12 ± 0.5 ^a^	0 ^b^	13 ± 1.0 ^a^	19 ± 1.0 ^a^	10 ±0.3 ^a^	14 ± 0.2 ^a^
*A. alternata*		13 ± 0.6 ^a^	14 ± 0.5 ^a^	0 ^b^	12 ± 0.3 ^a^	13 ± 0.5 ^a^	18 ±1.0 ^a^	12 ± 0.6 ^a^
*A. niger*		12 ± 1.0 ^a^	9 ± 0.5 ^a^	12 ± 0.5 ^a^	0 ^b^	0 ^b^	13 ± 0.6 ^a^	15 ± 0.5 ^a^

Values are given as mean ± S.D. of the triplicate experiment. ^(a)^ IZ: Diameter of inhibition zones including diameter of well 6 mm. ^(b)^ 0: No antimicrobial activity. ^(c)^ The concentration of Gentamicin used was 10 μg/well. ^(d)^ The concentration of Amphotericin B used was 20 μg/well. EtOAc: ethyl acetate; MeOH: methanol. Microorganisms: *B. subtilis: Bacillus subtilis* (ATCC 6633); *S. aureus: Staphylococcus aureus* (ATCC 25923); *S. epidermidis: Staphylococcus epidermis* (ATCC 12228); *E. faecalis: Enterococcus faecalis* (ATCC 29212); *M. luteus: Micrococcus luteus* (ATCC 1880); *E. coli: Escherichia coli (ATCC 25922); P. aeruginosa: Pseudomonas aeruginosa* (ATCC 9027); *K. pneumoniae: Klebsiella pneumoniae (*ATCC 10031*); S. enteritidis; Salmonella enteritidis* (CTM 2133). Fungal strains: *A. alternata; Alternaria alternata* (CTM10230); *F. oxysporum; Fusarium oxysporum* (CTM10402); *B. cinerea; Botrytis cinerea* (LBPES15); *A. niger: Aspergillus niger* (CTM 10099).

**Table 6 molecules-28-03947-t006:** Antimicrobial activity of the EtOAc (spring) against foodborne, spoiling bacteria and determination of its minimum inhibitory concentrations (MIC) and minimum bactericidal/fungicidal concentrations (MBC or MFC) expressed in mg·mL^−1^.

Microbial Strains	*A. henoniana* EtOAc Extract (mg·mL^−1^)			Gentamicin ^(c)^ (µg·mL^−1^)		
	MIC ^(a)^	MBC ^(b)^	MBC/MIC	MIC ^(a)^	MBC ^(b)^	MBC/MIC
**Gram-positive**						
*B* *. subtilis*	1.25	1.25	1	2.5	5	2
*S* *. aureus*	1.25	1.25	1	2.5	5	2
*S* *. epidermis*	0.625	1.25	2	2.5	10	4
*E* *. faecalis*	2.5	5	2	10	20	2
*M* *. luteus*	>10	>10	-	2.5	5	2
**Gram-negative**						
*E* *. coli*	-	-	-	2.5	5	2
*P* *. aeruginosa*	-	-	-	5	20	4
*K* *. pneumoniae*	5	5	1	5	10	2
*S*. *enteritidis*	10	10	1	10	>20	>2
		**Fungi**			**Amphotericin B ^(d)^** **(µg·mL^−1^)**	
	MIC ^(a)^	MFC ^(b)^	MFC/MIC	MIC ^(a)^	MFC ^(b)^	MFC/MIC
*F. oxysporum*	1.125	4.5	4	0.625	1.25	2
*A. alternata*	2.25	9	4	0.156	0.625	4

Values are given as mean ± S.D. of the triplicate experiment. EtOAc: ethyl acetate. ^(a)^ Minimum inhibitory concentrations (MIC) are expressed in mg·mL^−^^1^. ^(b)^ Minimum bactericidal concentrations (MBC) or minimum fungicidal concentration (MFC). ^(c)^ The concentration of Gentamicin used was 10 μg/well. ^(d)^ The concentration of Amphotericin B used was 20 μg/well. -: Not determined.

**Table 7 molecules-28-03947-t007:** Correlation among phenolic contents and inhibition zone diameter (mm) *.

Variables	TPC	TFC	*B*. *subtilis*	*S*. *aureus*	*S*. *epidermis*	*E*. *faecalis*	*M*. *luteus*	*K*. *pneumoniae*	*S* *. enteritidis*	*F. oxysporum*
TPC	1									
TFC	0.200	1								
*B* *. subtilis*	0.908 *	−0.543	1							
*S* *. aureus*	0.327	−0.982 *	0.693 *	1						
*S* *. epidermis*	−0.693 *	0.812 *	−0.932 *	−0.908 *	1					
*E* *. faecalis*	0.277	−0.991 *	0.655 *	−0.999 *	−0.885 *	1				
*M* *. luteus*	−0.786 *	−0.500	−0.454	0.327	0.099	0.376	1			
*K* *. pneumoniae*	−0.982 *	0.327	−0.971 *	−0.500	0.817 *	−0.454	0.655 *	1		
*S* *. enteritidis*	−0.155	−0.967 *	0.312	0.901 *	−0.636 *	0.923 *	0.705 *	−0.075	1	
*F. oxysporum*	−0.277	0.991 *	−0.655	−0.999 *	0.885 *	−0.999 *	−0.376	−0.454	−0.923 *	1

TPC: total phenols content; TFC: total flavonoid contents; *: data show the Pearson correlation coefficients (R^2^) between the parameters (*p* < 0.05).

**Table 8 molecules-28-03947-t008:** LC-MS/MS identification of ethyl acetate stem extracts.

N	T_R_ (min) Winter/Spring	UV (nm)	Area (%)		*m/z*	MS^2^	MS^3^	Attribution	Ref.
			Winter	Spring					
**1**	9.00/9.89	242–294	0.1	0.1	593	383/353(100)	575/503/473(100)/441/383/353	Vicenin-2 (Apigenin-6,8-di-*C*-glucoside)	[18]
**2**	10.47/11.30	282–382	0.4	0.2	563	545/503/473/443(100)/383/353		Apigenin-6-*C*-glucoside-8-*C*-arabinoside (schaftoside)	[19]
**3**	11.14/12.46	262	0.2	0.1	533	515/473/443(100)/383/353		Apigenin-*C*-pentoside-*C*-pentoside	[20]
**4**	-/12.49	266–326	-	0.3	769	623	357/315/300/271	Isorhamnetin glucosyl-di-rhamnoside	[21]
**5**	-/13.04	258–294	-	0.1	623	357/315(100)/300/271		Isorhamnetin-3-*O*-rutinoside	[22]
**6**	14.33/15.01	342–382	0.5	0.5	373	355(100)/337/301/263/151		7-Hydroxymatairesinol	[23]
**7**	14.66/15.27	282	1.0	1.5	473	455(100)/437/397/385/369/379/337	437(100)/419/379/367/165	Asiatic acid/madecassic acid derivative	[24]
**8**	15.92/16.51	274	0.7	1.0	399	381(100)/341/327		2-(2,6-Dihydroxy-4-methoxyphenyl)-4- hydroxy-3-(hydroxymethyl)-5,6- methoxy-6,5-(3-methylbut-2-enyl)- benzofuran	[25]
**9**	16.18/16.76	286–350	1.1	1.0	457	439(100)/421/381		Lucidenic acid A	[26]
**10**	16.45/16.95	270–350	4.0	2.2	371	353(100)/341//299/165		Caffeoyl glucarate (isomers)	[27]
**11**	17.07/17.58	338	1.9	1.3	455	437/385 (100)	367(100)/313/165/150	Unidentified	
**12**	17.55/18.12	282	1.4	1.8	357	339/285 (100)/151 /109		Unidentified	
**13**	19.00/19.12	262–346	3.4	1.2	313	298 (100)/269		Hedysarimpterocarpene A (HPA)/ Wedelolactone	[25]
**14**	19.27/19.80	274–318	2.6	2.5	355	337(100)	305/165(100)/150/136/108	Chebulic acid	[28]
**15**	20.51/20.59	270	2.3	1.3	439	421(100)/369/351		Prenylated licoriphenone	[29]
**16**	21.25/21.78	274	0.9	1.4	341	323 (100)/297/269/151		Caffeic acid-*O*-glycoside.	[30]
**17**	21.84/22.24	282–358	2.0	1.0	441	423(100)/371		Unidentified	
**18**	22.07/22.54	286–382	1.8	2.0	425	407/355(100)		Unidentified	
**19**	22.52/22.94	282	2.5	1.6	439	369(100)		Unidentified	
**20**	24.80/-	286–382	1.6	-	425	407/247/235(100)/217/165		Unidentified	
**21**	25.41/25.72	274	1.4	1.1	439	421(100)/367/165		Unidentified	
**22**	-/25.93	270–302	-	1.0	339	307/269/165(100)/150/137		Unidentified	
**23**	26.44/26.84	270–362	2.2	2.7	439	421(100)/367/165		Unidentified	
**24**	27.09/27.40	270–342	4.0	2.0	425	407(100)/355	389(100)/335/217/151	Unidentified	
**25**	-/29.13	278	-	3.1	425	407/353(100)/219/151		Unidentified	
**26**	29.30/29.60	282	1.9	1.1	423	405/353(100)/233		Unidentified	
**27**	29.63/29.97	282–362	4.3	4.0	425	407/353/219(100)177		Abscisic acid hexoside	[31]
**28**	30.57/30.99	270–302	1.0	0.8	421	403/351(100)/165		6,8-Diprenylkaempferol	[32]
**29**	31.35/31.40	278	1.7	2.8	423	405(100)/365		Sophoraflavanone G	[33]
**30**	32.08/32.36	270	4.5	4.8	423	405/353(100)/165		Unidentified	
**31**	32.55/32.87	270	5.4	5.9	423	405(100)		Sophoraflavanone G	[33]
**32**	34.31/34.52	282	3.0	2.4	409	231/219(100)/151		Unidentified	
**33**	34.74/35.09	282	2.1	1.3	409	391/385/235(100)/217/165		Unidentified	
**34**	35.62/35.91	270–366	6.2	8.2	409	391(100)/337/219/151		Kanzonol Y	[34]
**35**	36.70/36.95	286–358	2.5	1.2	407	389/337/219(100)/187		Unidentified	
**36**	37.98/38.00	266–370	3.9	4.3	407	337/271/233(100)/205		Unidentified	
**37**	38.48/38.75	270	3.6	6.7	407	375/165 (100)/150		Unidentified	
**38**	41.63/41.81	278	0.6	1.2	393	151(100)		Unidentified	
**39**	42.26/42.47	262–374	2.7	2.9	203	185/175/159/148(100)		Unidentified	
**40**	43.19/43.39	286	12.2	14.7	393	219(100)		Unidentified	
**41**	45.30/45.37	370	4.1	4.1	391	203 (100)/187/159		Hispaglabridin A	[35]
**42**	46.83/46.77	266–298	2.3	1.9	405	343/165(100)/136		Unidentified	
**43**	50.00/49.62	274	1.3	1.1	391	151(100)/219		Unidentified	
**44**	52.22/51.60	282–370	0.4	0.1	389	371/203(100)/185		Unidentified	
**45**	53.15/52.47	286–374	0.4	0.3	389	371/253(100)/135		Unidentified	

## Data Availability

Not applicable.

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
