# Peer review of "Effect of the Harvest Season of Anthyllis henoniana Stems on Antioxidant and Antimicrobial Activities: Phytochemical Profiling of Their Ethyl Acetate Extracts"

_molecules, 2023, doi:10.3390/molecules28093947_

Round 1

Reviewer 1 Report (Previous Reviewer 2)

Authors of manuscript evaluated biological activity of methanolic, hexane, ethyl acetate extracts of Anthyllis henoniana stems harvested in two different periods of time (February and May 2021). They determined antioxidant activity using four assays (DPPH, ABTS, FRAP and TAC) and antimicrobial activity of all extracts (zone inhibition), and additionally the MIC/MFC and MBC/MFC values for the ethyl acetate extracts. Furthermore, they identified about half compounds found in the ethyl acetate extracts using LC-MS chromatography.

I have w few comments:

There is no need to provide a citation after each sentence if it comes from the same publication, it is enough after the paragraph to which the citation relates, e.g. lines 39-41, in these two sentences is a twice citation from the same publication; it is enough to put the [3] after second sentence, in line 41. Similar situation in next sentences and also in whole introduction section.

Line 99: “In this study, three complimentary methods were used to evaluate the antioxidant activity of A. henoniana stems extracts harvested in two periods.” – Authors described four methods – DPPH, ABTS, FRAP and TAC.

Line 102, 440 and 437: should be “2,2-diphenyl-1-picrylhydrazyl”

Line 128: should be “DPPH” instead of ”DDPH”

Line 205: according to MDPI standard, “in vitro” should be not italic

Line 269: after “minimum bactericidal/fungicidal concentration” it should be “(MBC or MFC)”, because Authors determined minimal bactericidal concentration (MBC) and minimal fungicidal concentrations (MFC)

Line 272-273 and 274-275: These two sentences: “If the ratio is higher than 4, the extract is bacteriostatic.” and “If the ratio is more than four it is considered  as bacteriostatic or fungistatic.” are identical. Please improve it.

Table 7, second compound: please put the whole name of compound, not a short name “Apigenin-6-C-glu-8-C-ara”

Line 363: “another chalcone known as dihydrochalcone”, chalcones and dihydrochalcones are a different classes of flavonoid compounds; chalcones possess C=C bond and dihydrochalcones are derivatives/analogues of chalcones, because they have saturated C-C bond instead of C=C.

Lines 435-438: “The divergence of antioxidant capacity could not fully be described by a single method. Thus, it should be determined by several assays using the same initial concentration such as free radical scavenging 2,2′-diphenyl picrylhydrazyl (DPPH), ferric reducing power  (FRAP) and total antioxidant capacity (TAC).” – Authors forgot about ABTS assay.

Furthermore, the major compound in ethyl acetate extract is still unknown. It's a pity, and it definitely diminishes the value of the work.

In my opinion, manuscript should be corrected by a native speaker.

Author Response

Reviewer 1:

There is no need to provide a citation after each sentence if it comes from the same publication, it is enough after the paragraph to which the citation relates, e.g. lines 39-41, in these two sentences is a twice citation from the same publication; it is enough to put the [3] after second sentence, in line 41. Similar situation in next sentences and also in whole introduction section.

Response: Thank you for your suggestion, we changed it as requested.

Line 99: “In this study, three complimentary methods were used to evaluate the antioxidant activity of A. henoniana stems extracts harvested in two periods.” – Authors described four methods – DPPH, ABTS, FRAP and TAC.

Response: We apologize for this. We checked and changed it as requested.

Line 102, 440 and 437: should be “2,2-diphenyl-1-picrylhydrazyl”

Line 128: should be “DPPH” instead of ”DDPH”

Response: We apologize for these errors. We made the necessary corrections.

Line 205: according to MDPI standard, “in vitro” should be not italic

Response: We apologize for this. We checked and changed it as requested in the entire manuscript.

Line 269: after “minimum bactericidal/fungicidal concentration” it should be “(MBC or MFC)”, because Authors determined minimal bactericidal concentration (MBC) and minimal fungicidal concentrations (MFC).

Response: We apologize for this. We checked and changed it as requested.

Line 272-273 and 274-275: These two sentences: “If the ratio is higher than 4, the extract is bacteriostatic.” and “If the ratio is more than four it is considered as bacteriostatic or fungistatic.” are identical. Please improve it.

Response: Thank you for your suggestion. However, as we indicated in the manuscript “Antimicrobial substances are usually regarded as bactericidal or fungicidal if the MBC/MIC or MFC/MIC ratio is less or equal to four. If the ratio is higher than 4, the extract is bacteriostatic” there is a difference between the two used terms in this sentence (bactericidal and bacteriostatic). In fact, an extract is bacteriostatic when it prevents bacteria’s growth and it is addressed as bactericidal when it kills bacterias. This statement can be justified with the following research paper: Pankey, G. A., & Sabath, L. D. (2004). Clinical relevance of bacteriostatic versus bactericidal mechanisms of action in the treatment of Gram-positive bacterial infections. Clinical infectious diseases, 38(6), 864-870. DOI: 10.1086/381972.

Table 7, second compound: please put the whole name of compound, not a short name “Apigenin-6-C-glu-8-C-ara”

Response: Thank you for your suggestion, we changed it as requested.

Line 363: “another chalcone known as dihydrochalcone”, chalcones and dihydrochalcones are a different classes of flavonoid compounds; chalcones possess C=C bond and dihydrochalcones are derivatives/analogues of chalcones, because they have saturated C-C bond instead of C=C.

Response: We apologize for this. We checked and changed it to the compound name, which is Kanzonol Y.

Lines 435-438: “The divergence of antioxidant capacity could not fully be described by a single method. Thus, it should be determined by several assays using the same initial concentration such as free radical scavenging 2,2′-diphenyl picrylhydrazyl (DPPH), ferric reducing power  (FRAP) and total antioxidant capacity (TAC).” – Authors forgot about ABTS assay.

Response: We apologize for this. We checked and added ABTS as requested.

Furthermore, the major compound in ethyl acetate extract is still unknown. It's a pity, and it definitely diminishes the value of the work.

Response: We apologize for this. We agree with this statement. Indeed, it’s a pity that we couldn’t identify the major compound using LC-MS/MS analysis as there was no literature data that goes in good agreement with its fragmentation pattern. However, we are currently working on its isolation and the determination of its structure using NMR that will be the subject of future research study.

Comments on the Quality of English Language.

In my opinion, manuscript should be corrected by a native speaker.

Response: As requested, Cambridge proofreading checked and corrected the English of the manuscript (please find attached the certificate).

Reviewer 2 Report (New Reviewer)

Dear authors, 

The manuscript contains important data on effects of the harvest season of Anthyllis henoniana stems on the antioxidant and antimicrobial activities. However, it needs a major revision to improve the submission. My comments are as follows: 

1-Please justify the choice of such seasons;  only two (winter and spring) and not four seasons?

2-Chemical profiling should be done for different kinds of extracts, 

3-English review is highly recommended, e.g,  line 30, please change "component" to "compound" and so on throughout the text, 

4-Effects of harvest season on chemical profiling and associated biological activities in other plant species have to be documented in the Introduction section,

5-At the end of the Introduction section, the authors must indicate the gap of  knowledge before stating their objectives,

6- A combined analysis of variance must be done to better exploit the data (main factors, their interactions, etc). In this regard, the following references can be added and discussed with your results: 10.33263/BRIAC126.84418452; https://doi.org/10.1155/2023/6308773,

7-What kind of correlations did you run, (Pearson, Spearman) ?. Please indicate and justify such a choice?

8-  Multivariate statistical analysis can be done to better exploit the data (PCA, regressions, etc),

9-The Discussion section must be amended and the mechanisms behind the observed effects must be indicated in light of published literature. Please see the following references (10.33263/BRIAC126.84418452;  https://doi.org/10.1155/2023/6308773; 10.1016/j.bcab.2022.102569),

10- Pedoclimatic data of the sampling site must be provided in the Material and Methods section. 

Kind regards.  

English review is needed to improve the quality of writing. 

Author Response

Reviewer 2:

Comments and Suggestions for Authors

Dear authors, 

The manuscript contains important data on effects of the harvest season of Anthyllis henoniana stems on the antioxidant and antimicrobial activities. However, it needs a major revision to improve the submission. My comments are as follows: 

1-Please justify the choice of such seasons; only two (winter and spring) and not four seasons?

Response: Dear Reviewer, the choice of the harvest seasons was made because these two months correspond to the blooming and the dry state of A.henoniana. As mentioned by Louhaichi et al. the blooming phase of A.henoniana begins in late winter. Southern Tunisia has an arid Mediterranean climate with a long-term annual rainfall of 80 mm concentrated in the growing season between September–April and a dry season lasting about 4 months during May–August (references 4 and 5 in the manuscript). Therefore, we chose two harvest months: One where the plant is bloomed and the other one where the plant is dry. These two seasons appeared to be sufficient for the evaluation of A.henoniana stems.

2-Chemical profiling should be done for different kinds of extracts.

Response: Thank you for your suggestion. We understand that the chemical profiling of all obtained extracts is interesting. However, this will deviate us from the purpose of this study. As presented in the manuscript, both ethyl acetate extracts acted differently as antioxidants and antimicrobial. Therefore, we chosen to study their chemical composition to show how the detected compounds can affect the biological activities.

3-English review is highly recommended, e.g, line 30, please change "component" to "compound" and so on throughout the text,

Response: we changed it as requested and an English review by Cambridge proofreading was proceeded.

4-Effects of harvest season on chemical profiling and associated biological activities in other plant species have to be documented in the Introduction section,

Response: Thank you for your suggestion. We added it as requested (line 54-57).

5-At the end of the Introduction section, the authors must indicate the gap of  knowledge before stating their objectives,

Response: If I understood your statement, you want us to justify the aim of this study that was already indicated at the end of the introduction section: “Beside the antidiabetic and antioxidant activities of A. henoniana flowers, there are no specific scientific reports, nor specific references dealing with the antimicrobial activity and the identification of its extracts composition using chromatographic techniques.”

6- A combined analysis of variance must be done to better exploit the data (main factors, their interactions, etc). In this regard, the following references can be added and discussed with your results: 10.33263/BRIAC126.84418452; https://doi.org/10.1155/2023/6308773,

Response: Thank you again for your suggestion. We added statistical correlations (Pearson) between the phenolic content and the bacterial strains for better exploitation of the obtained data. We discussed the results using the following requested reference ( Ref: Zeroual, A.; Sakar, E. H.; Mahjoubi, F.; Chaouch, M.; Chaqroune, A.; Taleb, M.; Effects of extraction technique and solvent on phytochemicals, antioxidant, and antimicrobial activities of cultivated and wild rosemary (Rosmarinus officinalis L.) from taounate region (northern morocco). Biointerface Res. Appl. Chem. 2022, 12, 6, 8441–8452. Doi: 10.33263/BRIAC126.84418452.)

7-What kind of correlations did you run, (Pearson, Spearman) ?. Please indicate and justify such a choice?

Response: The correlation used in this study is Pearson in order to determine the linear correlation between two variables, which are in this case the total phenolic or flavonoid contents, and each antioxidant or antimicrobial assays. It is generally known that phenolic compounds are responsible for deactivating free radicals based on their ability to donate a hydrogen. It was also proven that they influence the antimicrobial effect. Different research studies reported a linear correlation of total phenolic and flavonoid content with antioxidant capacity as well as the antimicrobial effect (Ref: Quirós-Fallas, M. I., Vargas-Huertas, F., Quesada-Mora, S., Azofeifa-Cordero, G., Wilhelm-Romero, K., Vásquez-Castro, F.,& Navarro-Hoyos, M. (2022). Polyphenolic HRMS Characterization, Contents and Antioxidant Activity of Curcuma longa Rhizomes from Costa Rica. Antioxidants, 11(4), 620. Doi: 10.3390/antiox11040620 / Bouzayani, B.; Koubaa, I.; Frikha, D.; Samet, S.; Ben Younes, A.; Chawech, R.; Maalej, S.; Allouche, N.; Jarraya, R. Spectrometric analysis, phytoconstituents isolation and evaluation of in vitro antioxidant and antimicrobial activities of Tunisian Cistanche violacea (Desf). Chem. Pap. 2022, 76, 3031-3050. Doi: 10.1007/s11696-022-02082-7.)

8- Multivariate statistical analysis can be done to better exploit the data (PCA, regressions, etc),

Response: Thank you for your suggestion. The existing knowledge about A. henoniana is very limited therefore; we focused our main goal on the investigation of the chemical composition of its ethyl acetate extracts in order to provide an explanation for the obtained antioxidant and antimicrobial activities. We did not focus on the optimization, as it could be the next step for future research paper that focuses on a better data exploitation using other statistical tests (Principal component analysis PCA, regressions etc…). In this study, we determined the R2 coefficient, which is a statistical measure of how close the data is to the fitted regression line. Furthermore, we think that all statistical methods (PCA, regressions, etc) will lead to similar conclusions. Previous research papers used only the determination coefficient method (R2) that appears to be sufficient for the statistical analysis (Ref: Samet, S.; Ayachi, A.; Fourati, M.; Mallouli, L.; Allouche, N.; Treilhou, M.; Téné, N.; Mezghani-Jarraya, R. Antioxidant and antimicrobial activities of Erodium arborescens aerial part extracts and characterization by LC-HESI-MS2 of its acetone extract. Molecules. 2022, 27, 14, 4399. doi.org/10.3390/molecules27144399./Affes, S.; Ben Younes, A.; Frikha, D.; Allouche, N.; Treilhou, M.; Tene, N.; Jarraya, R. ESI-MS/MS analysis of phenolic compounds from Aeonium arboreum leaf extracts and evaluation of their antioxidant and antimicrobial activities. Molecules. 2021, 26, 14, 4338. Doi: 10.3390/molecules26144338).

9-The Discussion section must be amended and the mechanisms behind the observed effects must be indicated in light of published literature. Please see the following references (10.33263/BRIAC126.84418452; https://doi.org/10.1155/2023/6308773; 10.1016/j.bcab.2022.102569),

Response: Thank you for your suggestion. We added more discussion in the findings interpretation section using the suggested reference (Ref: Sakar, E. H.; Zeroual, A.; Kasrati, A.; Gharby, S. Combined Effects of Domestication and Extraction Technique on Essential Oil Yield, Chemical Profiling, and Antioxidant and Antimicrobial Activities of Rosemary (Rosmarinus officinalis L.). J. Food Biochem. 2023, 2023, 1–13. Doi: 10.1155/2023/6308773.)

10- Pedoclimatic data of the sampling site must be provided in the Material and Methods section. 

Response: Sorry for the lack of precision. We added it as requested line (435-438).

Comments on the Quality of English Language

English review is needed to improve the quality of writing. 

Response: As requested, Cambridge proofreading checked and corrected the English of the manuscript (please find attached the certificate).

Round 2

Reviewer 1 Report (Previous Reviewer 2)

Lines 42-45: The correction is wrong, it doesn't make sense. Please improve it.

“Line 272-273 and 274-275: These two sentences: “If the ratio is higher than 4, the extract is bacteriostatic.” and “If the ratio is more than four it is considered as bacteriostatic or fungistatic.” are identical. Please improve it.

Response: Thank you for your suggestion. However, as we indicated in the manuscript “Antimicrobial substances are usually regarded as bactericidal or fungicidal if the MBC/MIC or MFC/MIC ratio is less or equal to four. If the ratio is higher than 4, the extract is bacteriostatic” there is a difference between the two used terms in this sentence (bactericidal and bacteriostatic). In fact, an extract is bacteriostatic when it prevents bacteria’s growth and it is addressed as bactericidal when it kills bacterias. This statement can be justified with the following research paper: Pankey, G. A., & Sabath, L. D. (2004). Clinical relevance of bacteriostatic versus bactericidal mechanisms of action in the treatment of Gram-positive bacterial infections. Clinical infectious diseases, 38(6), 864-870. DOI: 10.1086/381972.”

In revised manuscript Lines 279-280 and 281-282: “If the ratio is higher  than 4, the extract is bacteriostatic.” And “If the ratio is more than four it is considered as bacteriostatic or fungistatic.” – in both sentences, which are very close to each other in the manuscript, the information is the same. Please choose one of them (maybe in lines 281-282) because this information in lines 279-280 is a repetition. I understand the difference between bactericidal and bacteriostatic, however Authors duplicate the same information in two different sentences.

Lines 383-384: Still is a mistake, in abstract Authors described that Kanzonol Y is a dihydrochalcone, and in lines 383-384 the same compound is chalcone. The correct sentence should be: “ Compound 34 (TR = 35.62 min) generated a molecular ion at m/z 409 and was attributed to another chalcone derivative  known as Kanzonol Y”. Instead of “chalcone derivative” it could be also “dihydrochalcone” as in abstract.

Author Response

Comments and Suggestions for Authors

Lines 42-45: The correction is wrong, it doesn't make sense. Please improve it.

Response: We sincerely apologize for these errors. Some sentences were mistakenly deleted in this version. We added them again (Line 41-45).

“Line 272-273 and 274-275: These two sentences: “If the ratio is higher than 4, the extract is bacteriostatic.” and “If the ratio is more than four it is considered as bacteriostatic or fungistatic.” are identical. Please improve it.

Response: Thank you for your suggestion. However, as we indicated in the manuscript “Antimicrobial substances are usually regarded as bactericidal or fungicidal if the MBC/MIC or MFC/MIC ratio is less or equal to four. If the ratio is higher than 4, the extract is bacteriostatic” there is a difference between the two used terms in this sentence (bactericidal and bacteriostatic). In fact, an extract is bacteriostatic when it prevents bacteria’s growth and it is addressed as bactericidal when it kills bacterias. This statement can be justified with the following research paper: Pankey, G. A., & Sabath, L. D. (2004). Clinical relevance of bacteriostatic versus bactericidal mechanisms of action in the treatment of Gram-positive bacterial infections. Clinical infectious diseases, 38(6), 864-870. DOI: 10.1086/381972.”

In revised manuscript Lines 279-280 and 281-282: “If the ratio is higher than 4, the extract is bacteriostatic.” And “If the ratio is more than four it is considered as bacteriostatic or fungistatic.” – in both sentences, which are very close to each other in the manuscript, the information is the same. Please choose one of them (maybe in lines 281-282) because this information in lines 279-280 is a repetition. I understand the difference between bactericidal and bacteriostatic, however Authors duplicate the same information in two different sentences.

Response: Again, we sincerely apologize for this. We checked and deleted the repetition as requested (Line 283-284).

Lines 383-384: Still is a mistake, in abstract Authors described that Kanzonol Y is a dihydrochalcone, and in lines 383-384 the same compound is chalcone. The correct sentence should be: “ Compound 34 (TR = 35.62 min) generated a molecular ion at m/z 409 and was attributed to another chalcone derivative  known as Kanzonol Y”. Instead of “chalcone derivative” it could be also “dihydrochalcone” as in abstract.

Response: Thank you for your suggestions, we apologize for these errors. We made the necessary corrections.

Reviewer 2 Report (New Reviewer)

Dear Authors, 

Thank you for addressing most of the comments raised. To my opinion, the manuscript can be accepted for publication in Molecules. 

Best regards. 

The manuscript requires English editing.  

Author Response

Dear Authors, 

Thank you for addressing most of the comments raised. To my opinion, the manuscript can be accepted for publication in Molecules. 

Best regards. 

Response: We sincerely want to thank you for reviewing the article and for providing us with remarks that helped us improve the quality of our manuscript.

Comments on the quality of english language

The manuscript requires English editing.  

Response: As requested, “Cambridge proofreading” checked and corrected the English of the manuscript (please find attached the certificate).

This manuscript is a resubmission of an earlier submission. The following is a list of the peer review reports and author responses from that submission.

Round 1

Reviewer 1 Report

Effect of the harvest period of Anthyllis henoniana stems on the antioxidant and antimicrobial activities: Phytochemical profiling of its ethyl acetate extracts

Taking into account the nature of the work, it seems more appropriate to take into account the development phase of the plant - not the date of harvesting the raw material.

It is difficult to talk about the highest activity of the raw material collected on one of the two dates.

If 20 compounds have been identified, it is worth indicating in the abstract not the groups of compounds, but which specific compounds dominated in the raw materials depending on the date of harvest (developmental stage of the plant).

Introduction. The first paragraph is so general that seems to be redundant. This part of the work has hardly been elaborated at all. Authors should familiarize themselves with the requirements of the publishing house and elaborate this part much more carefully once again.

After reviewing the results, I find the work to be inconsistent. There is no continuity of activities and of thought. Why was the raw material collected in two dates. It seems as if the results of analyzes from various laboratories were collected and compiled in a chaotic way into an article. There are no results regarding the content of identified compounds (HPLC analysis). Therefore, we do not know which specific compounds are responsible for biological activity of raw materials, which compounds dominate depending on the date of harvest (developmental stage) and the solvent used.

Information on the harvesting of the raw material and post-harvest treatment is scarce.

Considering the above, I do not recommend this paper for publication in the journal Molecules.

Reviewer 2 Report

Authors of manuscript analyzed methanolic, hexane, ethyl acetate extracts of Anthyllis henoniana stems harvested in two different periods of time (February and May 2021). Authors evaluated antioxidant and antimicrobial activity of these extracts.

In my opinion, the introduction section is really poor. Introduction based only on two articles, one of them is a self-citation of pervious work on the same plant, which were concern flowers extract of  Anthyllis henoniana, so the novelty of this manuscript is also low.

In manuscript there is no information why this strains of bacteria and fungi were selected to investigations ?

With regard antioxidant activity, Authors measured the activity using DPPH and FRAP method. The ABTS assay would give a broader spectrum of this activity.

Line 329: “chalcone known as dihydrochalcone”, I don’t understand this statement. Chalcones possess C=C bond and dihydrochalcone are a reduced form of chalcone (only C-C bond, without double bond); dihydrochalcones are another class of flavonoids, so the statement is wrong.

4.3.3. Total Antioxidant Capacity: please briefly explain the methodology.

Lines 441-442: I’m not sure if all selected strains of bacteria were grown in this short time.

Minor comments:

In whole manuscript the concentration unit it should be “mg·mL-1” instead of “mg.mL-1”.

Figure 1. On axis y it should be “OD” or “Optical density” or “Absorbance” instead of “D.O.”.

Line 168: “in vitro” should be italic

Line 196: should be “low inhibition zone”

Line 217: should be “B. cinerea

Table 5. should be “A. alternata

Figure 3 is unreadable, especially axis x and y.

Line 487: (5% ACN) – I think is should be deleted.

In literature 19, 22,24, 27 and 29 no or incomplete doi number.